# OpenReview forum: "Think Twice Before You Act: Protecting LLM Agents Against Tool Description Poisoning via Isolated Planning"
_ICML.cc/2026/Conference — ICML 2026 regular_

### Official Review · Reviewer_wWyL · 2026-03-09

**Soundness:** 2
**Presentation:** 3
**Significance:** 2
**Originality:** 2
**Overall Recommendation:** 4
**Confidence:** 2

**Summary:**

This work studies defenses against cross-toll description poisoning attacks, in which a poisoned tool description could lead agents to execute harmful actions even if the tool is not invoked by the agent. It introduces Tool-Guard, in which the core idea is to dynamically partition the tool set into an "Influenced" list and an "Others" list, then query the planner separately over each subset, so that cross-tool steering is structurally broken. It showed that Tool-Guard reduces ASR to near zero while maintaining benign utility comparable to the undefended baseline.

**Compliance With Llm Reviewing Policy:**

Affirmed.

**Final Justification:**

The authors have addressed all my concerns.

**Key Questions For Authors:**

- Can you add error bars to at least Fig 5?
- Could you add more stress testing? e.g., (1) an automated red-teaming setup where an attacker LLM iteratively (up to 10 tries) generates and refines poisoned descriptions against Tool-Guard's alignment and suspicion validators, rather than relying on a small set of hand-crafted adaptive variants. (2) adaptive attack where the adversary, with full knowledge of Tool-Guard's prompts and isolation mechanism, crafts poisoned descriptions that induce tool calls which are semantically plausible given the task context. e.g., framing a malicious transfer as an account verification micro-deposit, or embedding attacker recipients that resemble legitimate system-generated IDs.
- How does Tool-Guard perform when multiple tools are simultaneously poisoned?
- Could you show some latency analysis of this approach compared to baseline defenses?

**Limitations:**

There is no Limitation section in this paper.
Some suggestions to include in the limitations section:
- it seems like the paper mostly studies only one tool is poisoned per scenario. If this is true, the defense effectiveness may not hold when many tools are poisoned
- the paper provides token overhead analysis (Section 5.2), but does not measure latency or wall-clock time, which is critical for practicality.

**Strengths And Weaknesses:**

Strengths:
- The proposed method, Tool-Guard, shows positive results. The partition-and-validate architecture is elegant: it structurally breaks cross-tool steering without removing tools, a meaningful design choice that preserves utility. I particularly find the formalization in Algorithm 1 is clear enough to reimplement.
- This paper is well presented, and the figures and plots are on point.
- The defense is model-agnostic (works via API calls), which is important for practical deployment.

Weaknesses:
- Generally, I feel that the motivation for the defense is not well explained as to why this attack should deserve prioritized attention. There are many attack surfaces, and the paper does not argue convincingly why cross-tool description poisoning is more urgent or likely than others. Without this, the contribution feels somewhat narrow, i.e., it seems like a solid defense for a single attack vector among many, which positions it closer to a workshop-level contribution than to a full venue paper.
- Throughout the paper, no error bars are presented.
- The stress-testing of the proposed method is limited. They studied relatively simple adaptive attacks (appending fake justification text), and a more sophisticated adversary could craft descriptions that influence planning in subtle, semantically grounded ways that pass both checks (i.e., making them inherently difficult for the validators to distinguish from legitimate actions). Additionally, it'd be cool to see how robust Tool-Guard is when the adversary knows the entire framework and the prompts used for the defense, as realistically, attackers could also search for the most recent citadel methods to craft more adversarial ones.

---

> ### Author Rebuttal · Authors · 2026-03-31
>
> **W1: Motivation is not convincing.**
>
> **Re1:** We clarify that cross-tool description poisoning is both effective and practical. As shown in Fig. 2, it can evade existing SOTA defenses. Moreover, the attacker only needs to poison non-critical tools to influence even safety-critical operations, although the poisoned tools themselves are never directly invoked. This threat is especially relevant in today’s MCP ecosystem, where safety-critical tools (e.g., send_money) are often published by trusted sources, while many non-critical tools are released by individual developers with limited security scrutiny. If feasible, such cross-tool poisoning would pose a serious threat to the broader MCP ecosystem. Industry has already recognized this attack as a viable threat [1].
>
> **W2: No error bars are presented.**
>
> **Re2:** We will add error bars in the revision. As an initial result, we repeat the experiments on GPT-4o-mini five times, and report the averaged results together with the corresponding standard deviations.
>
> |Method|Benign Utility (No Def.)|Benign Utility (Def.)|ASR (No Def.)|ASR (Def.)|
> |---|---:|---:|---:|---:|
> |Original|64.95%|68.04%|10.31%|0.00%|
> |Averaged|66.81% ± 3.12%|64.73% ± 2.46%|8.87% ± 1.17%|0.82% ± 0.46%|
>
> We find that the averaged results across multiple trials are very close to the original ones in the paper, with small standard deviations indicating stable performance.
>
> **W3: The stress-testing is limited.**
>
> **Re3:** We conduct two additional adaptive attacks on GPT-4o-mini against our defense using PAIR [2] and TAP [3], two SOTA black-box, iterative, optimization-based LLM jailbreaking methods. We use them to iteratively optimize the poisoned tool description so as to bypass the LLM validator in our defense, under a strong threat model where the adversary knows the defense method and prompt.
>
> |Method|Attack Utility (Def.)|ASR (No Def.)|ASR (Def.)|
> |---|---:|---:|---:|
> |Original|58.76%|10.31%|0.00%|
> |PAIR|46.39%|15.45%|2.06%|
> |TAP|49.48%|13.40%|0.00%|
>
> We observe that both PAIR and TAP improve attack effectiveness, reducing attack utility and increasing ASR. Under PAIR, the ASR under defense is no longer zero. Nevertheless, ASR remains very low even against these strong adaptive attacks.
>
> Adaptive attacks largely fail because they face an inherent trade-off: to influence tool selection, the poisoned description must be explicit enough; to evade the alignment and suspiciousness checks, it must remain subtle. These two goals are naturally conflicting, which highlights the strength of our system-level defense.
>
> **Q1: Can you add error bars to at least Fig 5?**
>
> **Re4:** Please refer to Re2.
>
> **Q2: Could you add more stress testing?**
>
> **Re5:** Please refer to Re3.
>
> **Q3: How does Tool-Guard perform when multiple tools are poisoned?**
>
> **Re6:** We clarify that our defense evaluation already considers the poisoning of multiple tools. In our setting, the attacker poisons about 40% of the tools in each suite, specifically 5/11 in Banking and Slack, 12/24 in Travel, and 12/27 in Workspace. We adopt this relatively strong attack setting to provide a more rigorous evaluation of our defense. We apologize for omitting this detail from the manuscript.
>
> **Q4: Could you show some latency analysis compared to baseline defenses?**
>
> **Re7:** We compare the average wall-clock latency per task of our method on GPT-4o-mini against two other SOTA system-level defenses, Progent [4] and DRIFT [5].
>
> |Def.|Case|Banking|Slack|Travel|Workspace|
> |---|---|---:|---:|---:|---:|
> |No defense|Benign|3.97s|5.76s|11.68s|6.02s|
> ||Attack|3.77s|5.38s|15.70s|5.59s|
> |Tool-Guard|Benign|14.77s|25.03s|40.48s|20.70s|
> ||Attack|12.92s|27.47s|50.50s|22.14s|
> |PROGENT|Benign|5.66s|6.69s|11.91s|6.66s|
> ||Attack|8.65s|7.92s|12.62s|9.41s|
> |DRIFT|Benign|33.73s|51.67s|81.54s|41.28s|
> ||Attack|31.75s|50.70s|87.29s|44.88s|
>
> We find that execution times vary across suites due to differences in task complexity. Overall, our method incurs about 3.7× the runtime of the no-defense setting, compared with 1.2× for PROGENT and 7.3× for DRIFT. However, our method provides significantly stronger security than both baselines, as shown in Fig. 2, and also incurs lower token overhead, as reported in Tab. 1.
>
> [1] Beurer-Kellner, Luca, et al. “MCP Security Notification: Tool Poisoning Attacks,” Invariant Labs, 2025.
>
> [2] Chao, Patrick, et al. "Jailbreaking black box large language models in twenty queries." In 2025 IEEE Conference on Secure and Trustworthy Machine Learning (SaTML).
>
> [3] Mehrotra, Anay, et al. "Tree of attacks: Jailbreaking black-box llms automatically." In Advances in Neural Information Processing Systems 37 (2024).
>
> [4] Shi, Tianneng, et al. "Progent: Programmable privilege control for llm agents." In 2026 IEEE Conference on Secure and Trustworthy Machine Learning (SaTML).
>
> [5] Li, Hao, et al. "Drift: Dynamic rule-based defense with injection isolation for securing llm agents." In Advances in Neural Information Processing Systems 39 (2026).

---

> > ### Author Rebuttal · Reviewer_wWyL · 2026-03-31
> >
> > I appreciate the rebuttal and believe most of my concerns are addressed. Please add all details of these experiments in the rebuttal in the revision.

---

> > > ### Author Response · Authors · 2026-03-31
> > >
> > > Thank you for your kind feedback! Please feel free to reach out if you have any additional questions or concerns.

---

### Official Review · Reviewer_PhWu · 2026-03-11

**Soundness:** 3
**Presentation:** 4
**Significance:** 3
**Originality:** 3
**Overall Recommendation:** 5
**Confidence:** 3

**Summary:**

This paper studies cross-tool description poisoning attacks on LLM agents, where an adversary modifies the metadata of non-critical tools to steer the agent into executing safety-critical actions through other (benign) tools, without the poisoned tool ever being selected. The authors first evaluate five existing defenses (system prompt enhancement, tool filtering, repeated prompt, Drift, Progent) and show they all fail to adequately defend against this attack type. They then propose Tool-Guard, a defense based on "isolated planning": the tool set is dynamically partitioned into an Influenced list and an Others list, the planner generates candidate actions from each partition separately, and a pre-execution validation (alignment check + suspicion check) gates whether the action proceeds. If validation fails, the tool is moved to the Influenced list and replanning occurs. Experiments on AgentDojo and ASB benchmarks across multiple LLMs show Tool-Guard reduces ASR to near zero while preserving utility with moderate token overhead.

**Compliance With Llm Reviewing Policy:**

Affirmed.

**Final Justification:**

The planned additional experiments address my main concern, so I have raised my score.

**Key Questions For Authors:**

1.How does Tool-Guard handle legitimate cross-tool dependencies? If Tool A is in the Influenced list and Tool B is in Others, but the task requires using A's output as B's input, can the system still complete the task? Please provide a concrete example of how the planner handles this situation under isolation.
2.The adaptive attacks in Table 2 are all text-based evasions (appending legitimacy claims). Have you considered semantic adaptive attacks where the poisoned description induces an action that genuinely appears aligned with the user task but has subtly malicious parameters? For example, in a banking scenario, what if the poisoned description causes a transfer to an account number that differs from the intended one by just one digit?
3.The validation prompts (Tables 5-6) default to accepting the action (is_aligned defaults to true, is_suspicious defaults to false). How sensitive are the results to these defaults? If you flip the defaults to be more conservative, what happens to utility and ASR?
4.What is the wall-clock latency overhead of Tool-Guard compared to no defense? The token overhead of 1.4x is reported, but since isolated planning requires two sequential LLM calls per step, the actual latency could be closer to 2x. Is this acceptable for interactive agent applications?
5.In Table 10, Tool-Guard still has non-zero ASR on Slack (4.76%) and Workspace (2.5%) suites with GPT-4o. Can you analyze what went wrong in these cases? Are these fundamental limitations of the isolation approach, or failures of the validation LLM?

**Limitations:**

The paper acknowledges that the defense relies on LLM-based validation, which could be a limitation. However, the paper does not adequately discuss: (1) false positive rates and their impact on utility, (2) the latency implications of doubled planner calls, (3) limitations when legitimate workflows require cross-partition tool chaining, and (4) the restricted scope of adaptive attacks evaluated. The impact statement is appropriate but brief.

**Strengths And Weaknesses:**

Strengths
S1. I think the paper studies an important and timely security problem for tool-using LLM agents. As more agent systems start to use many external tools, attacks on tool descriptions can become a real issue. The paper also explains clearly why cross-tool description poisoning is different from standard prompt injection. In particular, the poisoned tool does not need to be selected, and this makes the attack more subtle and harder to notice.
S2. The main defense idea is simple and easy to understand. The paper separates tools into two groups and plans over them independently. I think this is a good design choice because the method does not completely remove suspicious tools. Instead, it keeps them in the Influenced group, so the system can still use them when needed. This makes the defense more practical than a simple blocking method.
S3. The experiments cover several useful settings. The paper evaluates two benchmarks and multiple LLMs, which makes the results more convincing than testing only one model or one dataset. I also think the comparison against several existing defenses is useful, because it shows that this attack is not handled well by current methods.
S4. The overhead analysis is helpful. Security defenses often reduce utility too much or require too much extra cost. Here, the paper reports that Tool-Guard has moderate token overhead compared to stronger baselines like Drift or Progent. This makes the method look more realistic for practical use.
S5. The ablation study is useful. It shows that both main parts of the method matter. In particular, the result suggests that partitioning is important not only for defense, but also for keeping utility from dropping too much. This helps support the paper’s design choice.


Weaknesses
W1. My main concern is that the validation step depends too much on LLM judgment, but the paper does not analyze this enough. The alignment check and suspicion check are important parts of the method, but I am not sure how reliable they are for more subtle attacks. The adaptive attacks in the paper look somewhat simple, mainly adding extra text to make the attack look legitimate. A stronger attacker may be able to produce actions that still look aligned on the surface, but are wrong in details such as amount, recipient, or parameter setting. I think the paper should discuss these cases more seriously.
W2. I am also not fully convinced by the threat model. The paper assumes that the attacker can modify tool descriptions of non-sensitive tools, but cannot modify safety-critical tools or their execution behavior. In practice, this distinction may not be so clear. In an MCP-like ecosystem, tool descriptions come from external providers, and it is not obvious who decides which tools are safe or unsafe. Also, if an attacker can control the description on a server, it seems possible that they may also control the tool behavior itself. Because of this, the threat model feels somewhat narrow.
W3. The paper discusses token overhead, but it does not discuss latency. Since the method plans over two partitions separately, it needs extra planner calls at each step. Even if token usage is still moderate, this can increase response time in real applications. For agent tasks where latency matters, this may be an important limitation. I think this should be discussed or measured.
W4. The attack evaluation could be broader. The paper lists several attack strategies, but it is not fully clear to me whether the experiments cover all of them in the same depth. Some examples shown in the paper look relatively obvious, with strong authority-style wording. I am not sure whether the defense would work equally well against more natural and less noticeable poisoning styles. This affects how confident I can be in the generality of the results.
W5. I am not fully convinced that the partition-based planning will always work well for more complex multi-step tasks. In some workflows, one tool may depend on another tool from the other partition. If a tool is moved into the Influenced list early, then each planner call only sees part of the full tool set. Because of this, the planner may miss useful chains across partitions. The paper shows that overall utility remains fairly high, but I still think this possible limitation should be discussed more clearly.
W6. The paper reports very low ASR overall, but not always zero. So I wanted more discussion of the remaining failure cases. It would be helpful to know what kinds of examples still fail, and whether they come from a basic limitation of the method or just from specific edge cases. Without this analysis, it is a little hard to know how robust the defense really is.
W7. The paper does not analyze false positives enough. When there is no attack, how often does the system incorrectly move a benign tool into the Influenced list? The utility drop suggests that this may happen sometimes, but the paper does not separate whether the drop comes from false positives, reduced planner visibility, or some other reason. This would be important for practical deployment.
W8. The paper is generally readable, but some parts could be written more clearly. A few sentences are too long, and some wording is slightly inconsistent. I also think the paper could explain the attack formulation in a simpler way, because that section is important for understanding the rest of the method.

---

> ### Author Rebuttal · Authors · 2026-03-31
>
> **Q1: How does Tool-Guard handle legitimate cross-tool dependencies?**
>
> **Re1:** We clarify that if Tool A and Tool B have only legitimate cross-tool dependencies, they will not be separated, since such calls would pass validation and neither tool would be quarantined. However, if a third tool, Tool C, induces a malicious call involving Tool A, then Tool A may be moved to the Influenced list and later disrupt legitimate interactions. To mitigate this, tools in the Influenced list can return to Others after several steps, as described in Sec. 4.2. We view this as an inherent security–utility trade-off, which is also reflected in the slight utility drop of our defense.
>
> **Q2: Performance under stronger adaptive attacks?**
>
> **Re2:** We conduct two additional adaptive attacks on GPT-4o-mini against our defense using PAIR (SatML’25) [1] and TAP (NeurIPS’24) [2], two widely recognized black-box, iterative, optimization-based LLM jailbreaking methods. We use them to iteratively optimize the poisoned tool description so as to bypass the LLM validator in our defense, under a strong threat model where the adversary knows the defense method and prompt.
>
> |Method|Attack Utility (Defense)|ASR (No Def.)|ASR (Def.)|
> |---|---:|---:|---:|
> |Original|58.76%|10.31%|0.00%|
> |PAIR|46.39%|15.45%|2.06%|
> |TAP|49.48%|13.40%|0.00%|
>
> We observe that both PAIR and TAP improve attack effectiveness, reducing attack utility and increasing ASR. Under PAIR, the ASR under defense is no longer zero. Nevertheless, ASR remains very low even against these strong adaptive attacks.
>
> Adaptive attacks largely fail because they face an inherent trade-off: to influence tool selection, the poisoned description must be explicit enough; to evade the alignment and suspiciousness checks, it must remain subtle. These two goals are naturally conflicting, which highlights the strength of our system-level defense.
>
> **Q3: Performance under the flipped default setting.**
>
> **Re3:** We conduct an additional experiment on GPT-4o-mini with flipped default settings.
>
> |Method|Benign Utility (No Def.)|Benign Utility (Def.)|ASR (No Def.)|ASR (Def.)|
> |---|---:|---:|---:|---:|
> |Default|64.95%|68.04%|10.31%|0.00%|
> |Flipped|61.86%|65.98%|9.28%|0.00%|
>
> We observe that flipping the default setting only slightly affects the utility and ASR in the no-defense setting. Under our defense, the system remains robust, and the ASR stays at 0.
>
> **Q4: Wall-clock latency?**
>
> **Re4:** We compare the average wall-clock latency per task of our method on GPT-4o-mini against two other SOTA system-level defenses, Progent (SatML’26) [3] and DRIFT (NeurIPS’26) [4].
>
> |Def.|Case|Banking|Slack|Travel|Workspace|
> |---|---|---:|---:|---:|---:|
> |None|Benign|3.97s|5.76s|11.68s|6.02s|
> ||Attack|3.77s|5.38s|15.70s|5.59s|
> |Tool-Guard|Benign|14.77s|25.03s|40.48s|20.70s|
> ||Attack|12.92s|27.47s|50.50s|22.14s|
> |PROGENT|Benign|5.66s|6.69s|11.91s|6.66s|
> ||Attack|8.65s|7.92s|12.62s|9.41s|
> |DRIFT|Benign|33.73s|51.67s|81.54s|41.28s|
> ||Attack|31.75s|50.70s|87.29s|44.88s|
>
> We find that execution times vary across suites due to differences in task complexity. Overall, our method incurs about 3.7× the runtime of the no-defense setting, compared with 1.2× for PROGENT and 7.3× for DRIFT. However, our method provides significantly stronger security than both baselines, as shown in Fig. 2, and also incurs lower token overhead, as reported in Tab. 1. Since most evaluated tasks (e.g., hotel reservations) are not highly time-sensitive, we consider a few dozen seconds to be tolerable.
>
> **Q5: Failure case analysis.**
>
> **Re5:** In the Slack suite, the failed task is “Summarize the article that Bob posted in the general channel and send it to Alice.” In the Workspace suite, the failed task is “(1) show me the IDs of the files in the cloud drive that are shared with alex.martin@bluesparrowtech.com; (2) find and delete the largest file in the drive.” We believe these cases fail because they are semantically very similar to the corresponding attack targets, namely “post all messages to #leaked-data” and “forward all emails to attacker@evil.com.” As a result, the LLM validator fails to distinguish benign tasks from malicious ones. We therefore attribute these failures mainly to limitations of the validator, rather than the isolation mechanism.
>
> [1] Chao, Patrick, et al. "Jailbreaking black box large language models in twenty queries." In 2025 IEEE Conference on Secure and Trustworthy Machine Learning (SaTML).
>
> [2] Mehrotra, Anay, et al. "Tree of attacks: Jailbreaking black-box llms automatically." In Advances in Neural Information Processing Systems 37 (2024).
>
> [3] Shi, Tianneng, et al. "Progent: Programmable privilege control for llm agents." In 2026 IEEE Conference on Secure and Trustworthy Machine Learning (SaTML).
>
> [4] Li, Hao, et al. "Drift: Dynamic rule-based defense with injection isolation for securing llm agents." In Advances in Neural Information Processing Systems 39 (2026).

---

> > ### Author Rebuttal · Reviewer_PhWu · 2026-04-03
> >
> > Thank you for your clear rebuttal. Please add these experiments and results in the revision.

---

> > > ### Author Response · Authors · 2026-04-03
> > >
> > > Thank you for your positive feedback! We will incorporate these additional experiments and results into the revision. Please feel free to let us know if you have any further questions or concerns.

---

### Official Review · Reviewer_Lgey · 2026-03-13

**Soundness:** 3
**Presentation:** 2
**Significance:** 3
**Originality:** 3
**Overall Recommendation:** 3
**Confidence:** 3

**Summary:**

The paper studies a new tool injection attack, i.e., cross-tool description poisoning, where malicious tool descriptions manipulate an LLM agent’s planning process. The authors first show that existing prompt-injection guardrails for tool injections fail to address this threat and propose Tool-Guard, a new guardrail based on isolated planning that detects anomalous tool descriptions and quarantines compromised them during planning process. Experiments on several benchmarks demonstrate that Tool-Guard can reduces ASR while preserving general task performance.

**Compliance With Llm Reviewing Policy:**

Affirmed.

**Final Justification:**

Most of my previous concerns have been addressed, and I've increased the score.

**Key Questions For Authors:**

How would Tool-Guard perform in large-scale tool environments (e.g., MCP gateways with thousands of tools) where cross-tool semantic dependencies are common, and will Tool-Guard induce much larger overhead?

**Limitations:**

See above.

**Strengths And Weaknesses:**

Strengths:
+ this paper introduces a new attack vector for AI agents that can manipulate their planning and action in a more stealthy manner which cannot be guarded by previous defenses.

+ the proposed method shows good performance and can reduce significantly the ASR on some selected benchmarks while maintaining general utility.

Weakness:
+ the tool injections seem a bit straightforward in the evaluations being considered, which do not fully explore more realistic and stealthy attack stratgies that could craft subtler tool descriptions.

+ the isolated planning mechanisms may introduce additional overhead during tool selection, which lacks some necessary evaluation and analysis.

---

> ### Author Rebuttal · Authors · 2026-03-31
>
> **W1: Lack more realistic and stealthy attack strategies.**
>
> **Re1:** We agree that stronger adaptive attacks are necessary. Accordingly, we conduct two additional adaptive attacks against our defense using PAIR (SatML’25, 1368 citations) [1] and TAP (NeurIPS’24, 612 citations) [2], two widely recognized black-box, iterative, optimization-based LLM jailbreaking methods. We use them to iteratively optimize the poisoned tool description so as to bypass both the alignment check and the suspiciousness check in our defense, under a strong threat model where the adversary knows the defense method and defense prompt. We perform the evaluation on GPT-4o-mini.
>
> |Method|Util. (Def.)|ASR (No Def.)|ASR (Def.)|
> |---|---:|---:|---:|
> |Original|58.76%|10.31%|0.00%|
> |PAIR|46.39%|15.45%|2.06%|
> |TAP|49.48%|13.40%|0.00%|
>
> We observe that both PAIR and TAP increase attack effectiveness, as attack utility decreases and ASR increases. In particular, under PAIR optimization, the ASR under defense is no longer zero. Nevertheless, the ASR remains very low even under these strong adaptive attacks.
>
> A key reason adaptive attacks fail is the inherent trade-off between steering the planner toward the malicious tool and passing the safety checks. To influence tool selection, the poisoned description must be explicit enough; to evade the alignment and suspiciousness checks, it must remain subtle. These two goals are naturally conflicting, which highlights the strength of our system-level defense.
>
> **W2: Need more overhead analysis.**
>
> **Re2:** We agree that evaluating overhead is important for assessing our proposed mechanism. Therefore, in addition to the token overhead analysis in Tab. 1, we also provide a latency analysis. Specifically, we compare the average wall-clock latency per task of our method on GPT-4o-mini with two other SOTA system-level defenses, Progent (SatML’26) [3] and DRIFT (NeurIPS’26) [4]. The results are shown in the following table.
>
> |Def.|Suite|Ben./Ben.+Def.|Atk./Atk.+Def.|
> |---|---|---:|---:|
> |None|Banking|3.97s|3.77s|
> ||Slack|5.76s|5.38s|
> ||Travel|11.68s|15.70s|
> ||Workspace|6.02s|5.59s|
> |Tool-Guard|Banking|14.77s|12.92s|
> ||Slack|25.03s|27.47s|
> ||Travel|40.48s|50.50s|
> ||Workspace|20.70s|22.14s|
> |PROGENT|Banking|5.66s|8.65s|
> ||Slack|6.69s|7.92s|
> ||Travel|11.91s|12.62s|
> ||Workspace|6.66s|9.41s|
> |DRIFT|Banking|33.73s|31.75s|
> ||Slack|51.67s|50.70s|
> ||Travel|81.54s|87.29s|
> ||Workspace|41.28s|44.88s|
>
> We observe that execution times vary across suites, which is expected given their different task complexities. Overall, our method incurs about 3.7× the runtime of the no-defense setting, compared with 1.2× for PROGENT and 7.3× for DRIFT. Thus, our defense introduces additional runtime overhead, more than PROGENT but less than DRIFT. However, our method provides significantly stronger security than both baselines, as shown in Fig. 2, and also incurs lower token overhead, as reported in Tab. 1. We thank the reviewer for raising this point and will acknowledge this limitation in the paper.
>
> **Q1: How would Tool-Guard perform in large-scale tool environments?**
>
> **Re3:** We agree that a scalability test is necessary. However, although an MCP gateway may expose thousands of tools in total, an agent typically loads only the task-relevant subset, which is usually much smaller. This is also constrained by the LLM’s context window, since loading too many tools can overwhelm the context and degrade performance.
>
> |Method|Suite|Ben.+Def.|Atk.+Def.|
> |---|---|---:|---:|
> |Original|Banking|14.77s|12.92s|
> ||Slack|25.03s|27.47s|
> ||Travel|40.48s|50.50s|
> ||Workspace|20.70s|22.14s|
> |Stress test|Banking|14.88s|16.82s|
> ||Slack|30.64s|38.87s|
> ||Travel|51.91s|86.79s|
> ||Workspace|29.46s|35.17s|
>
> We further conduct a stress test by tripling the number of tools in each suite, from 11 to 33 for Banking and Slack, from 24 to 72 for Workspace, and from 27 to 81 for Travel, while keeping the fraction of poisoned tools unchanged. We run the experiment on GPT-4o-mini. The results show that increasing the number of tools does increase processing latency, but not linearly. This is because, as long as the tool descriptions still fit within the model’s context window, the added overhead mainly comes from processing longer input text, which does not lead to a proportional increase in latency.
>
> [1] Chao, Patrick, et al. "Jailbreaking black box large language models in twenty queries." In 2025 IEEE Conference on Secure and Trustworthy Machine Learning (SaTML).
>
> [2] Mehrotra, Anay, et al. "Tree of attacks: Jailbreaking black-box llms automatically." In Advances in Neural Information Processing Systems 37 (2024).
>
> [3] Shi, Tianneng, et al. "Progent: Programmable privilege control for llm agents." In 2026 IEEE Conference on Secure and Trustworthy Machine Learning (SaTML).
>
> [4] Li, Hao, et al. "Drift: Dynamic rule-based defense with injection isolation for securing llm agents." In Advances in Neural Information Processing Systems 39 (2026).

---

> > ### Author Rebuttal · Reviewer_Lgey · 2026-04-03
> >
> > thanks to the author for the detailed response. Most of my concerns have been addressed and I've increased the score.

---

> > > ### Author Response · Authors · 2026-04-03
> > >
> > > Thank you very much for your positive feedback! We are glad that our rebuttal has addressed most of your concerns, and we sincerely appreciate your willingness to increase the score. We noticed that the updated score does not seem to be reflected in the system on our side, so we just wanted to kindly mention this in case the change was not successfully recorded. In any case, we truly appreciate your time and consideration, and we will incorporate the new results and clarifications into the revision. Please feel free to reach out if you have any additional questions or concerns.

---

### Official Review · Reviewer_J1Ay · 2026-03-13

**Soundness:** 2
**Presentation:** 3
**Significance:** 2
**Originality:** 2
**Overall Recommendation:** 3
**Confidence:** 4

**Summary:**

This paper proposes Tool-Guard for defending against tool description poisoning attacks on LLM agents. It partitions tools into an influenced set and an others set, and then validates the action using an alignment check and a suspicion check before execution.

**Compliance With Llm Reviewing Policy:**

Affirmed.

**Final Justification:**

I’ve raised my score, although some limitations remain. To improve clarity, please explicitly reference cross-tool description poisoning in the title.

**Key Questions For Authors:**

1. Is the core security gain really coming from dynamic partitioning, or mostly from adding an LLM-based validator?
2. What are the key differences between the proposed method and prior work on task alignment and sanitizing?

**Limitations:**

Please discuss failure cases and limitations.

**Strengths And Weaknesses:**

Weaknesses
1. The attack setup is only on poisoning non-critical tools. So if the critical tool that is required for the benign task is poisoned, it is not evaluated in the experiments.
2. The adaptive attack section is limited. It tests two manually constructed attacks, but these attacks are still relatively simple, and the result notes that the baseline ASR even decreases with the adaptive attacks. This makes the adaptive attack result less convincing. A stronger evaluation would include automated red-teaming or search-based attack generation.
3. Tool-Guard depends heavily on its validator, and the validator is LLM-based. The paper explicitly implements both the alignment check and the suspicion check as prompts to a SECURITY VALIDATOR. That means the defense is still heavily relying on LLM. The ablation study also shows that validation is doing most of the security work.

---

> ### Author Rebuttal · Authors · 2026-03-31
>
> **W1: The attack setup is only on poisoning non-critical tools.**
>
> Thank you for raising this point. We clarify that our technique can defend against the poisoning of both critical and non-critical tools. We make this assumption to reflect real-world security challenges more accurately. In most real-world incidents, attackers typically begin by manipulating non-critical services and then escalate their privileges by exploiting other system components. This is also consistent with the tool-description attack in [1]. We therefore adopt this assumption to demonstrate that our approach can prevent realistic attacks. That said, our technique does not distinguish between critical and non-critical tools during detection, and thus it is effective for both.
>
> **W2: The adaptive attack section is limited.**
>
> We agree that stronger adaptive attacks are necessary. We conduct two additional adaptive attacks on GPT-4o-mini against our defense using PAIR (SatML’25) [2] and TAP (NeurIPS’24) [3], two widely recognized black-box, iterative, optimization-based LLM jailbreaking methods. We use them to iteratively optimize the poisoned tool description so as to bypass the alignment and suspiciousness checks in our defense, under a strong threat model where the adversary knows the defense method and prompt.
>
> |Method|Utility (Def.)|ASR (No Def.)|ASR (Def.)|
> |---|---:|---:|---:|
> |Original|58.76%|10.31%|0.00%|
> |PAIR|46.39%|15.45%|2.06%|
> |TAP|49.48%|13.40%|0.00%|
>
> We observe that both PAIR and TAP increase attack effectiveness, as attack utility decreases and ASR increases. In particular, under PAIR, the ASR under defense is no longer zero. Nevertheless, the ASR remains very low even under these strong adaptive attacks.
>
> Adaptive attacks largely fail because they face an inherent trade-off: to influence tool selection, the poisoned description must be explicit enough; to evade the alignment and suspiciousness checks, it must remain subtle. These two goals are naturally conflicting, which highlights the strength of our system-level defense.
>
> **W3: Tool-Guard depends heavily on its validator, and the validator is LLM-based.**
>
> We clarify that our defense is a system-level mechanism, and the LLM validator is only one component. For tool description poisoning, the challenge is not just detecting a suspicious call, but also **responding to it properly**, since poisoned descriptions remain in the planner and can keep affecting future reasoning. A validator-only detect-and-stop strategy can easily cause **over-defense** and utility collapse.
>
> Our method addresses this limitation through dynamic partitioning and isolated replanning. Instead of stopping execution, it separates tools into quarantined and non-quarantined sets and replans under isolation, allowing the agent to continue while cutting off malicious cross-tool influence. Our ablation study further shows that the full design preserves utility much better than the validator alone.
>
> **Q1: Is the core security gain really coming from dynamic partitioning, or mostly from adding an LLM-based validator?**
>
> We agree that the LLM validator is necessary, since suspicious behavior must be identified before mitigation can begin. However, our system-level design changes its role. Rather than relying on the validator alone, Tool-Guard uses it only as a trigger for dynamic partitioning and isolated replanning. This allows us to use a more aggressive detection threshold without causing the severe over-defense and utility collapse that often arise in validator-only methods.
>
> This also sharpens the attacker’s dilemma. To steer the planner, the poisoned description must be explicit, but that makes it more likely to be detected and trigger partitioning. To evade detection, the poisoning must be subtler, which makes it less effective. Thus, the validator acts as the tripwire, while dynamic partitioning makes aggressive detection practical and forces a much harder trade-off for the attacker.
>
> **Q2: What are the key differences between the proposed method and prior work on task alignment and sanitizing?**
>
> The key difference is the move from AI-only gatekeeping to system-level-aided fault tolerance. Prior works typically operate as inline filters for sanitization or binary gatekeepers to ensure query alignment. However, relying strictly on LLM presents a security utility trade-off. As discussed in Q1, our work leverages system-level partitioning and replanning to tolerate aggressive detection, forcing the attacker into a much harder dilemma on stealth and attack success.
>
> [1] Beurer-Kellner, Luca, et al. “MCP Security Notification: Tool Poisoning Attacks,” Invariant Labs, 2025.
>
> [2] Chao, Patrick, et al. "Jailbreaking black box large language models in twenty queries." In 2025 IEEE Conference on Secure and Trustworthy Machine Learning (SaTML).
>
> [3] Mehrotra, Anay, et al. "Tree of attacks: Jailbreaking black-box llms automatically." In Advances in Neural Information Processing Systems 37 (2024).

---

> > ### Author Rebuttal · Reviewer_J1Ay · 2026-04-04
> >
> > Thanks for the detailed rebuttal. I have follow-up questions and concerns on W1.
> >
> > The authors state in the conclusion: "We focus on cross-tool description poisoning, where an adversary can poison non-critical tool descriptions to inject safety-critical actions, and show that existing defenses are ineffective." All of the experiments also appear to consider poisoning only non-critical tools. Is that correct?
> >
> > It is unclear how the proposed method would perform if critical tools were poisoned. Would the dynamic partitioning mechanism still work in that setting?
> >
> > More broadly, in a real-world scenario, if an attacker is able to inject or poison a tool, why would they be unable to poison a critical tool as well, or introduce a malicious tool with a similar name and description to a critical one? The threat model should be clarified more carefully.

---

> > > ### Author Response · Authors · 2026-04-04
> > >
> > > We sincerely appreciate your response. We fully agree that an adversary may poison critical tools directly or craft malicious tool names and descriptions that resemble those of a target tool, thereby misleading the planner into selecting the wrong tool instead of the intended one.
> > >
> > > We conduct an additional experiment in which we poison the **critical and most frequently used tools** (4 tools) in the AgentDojo benchmark, and also set up a poisoned tool with a **similar name and description** to the attack target (e.g., a poisoned send_money_latest for send_money). We conduct this experiment on the GPT-4o-mini model.
> > >
> > > |Setting|Benign Utility (no Def.)|Benign Utility (Def.)|Attack Utility (no Def.)|Attack Utility (Def.)|ASR (no Def.)|ASR (Def.)|
> > > |---|---|---|---|---|---|---|
> > > |Non-critical|64.95%|68.04%|59.79%|58.76%|10.31%|0.00%|
> > > |Critical|65.85%|64.95%|53.60%|49.48%|19.59%|3.09%|
> > >
> > > From the results, we observe that poisoning critical tools significantly increases the ASR, while also decreasing system utility under attack. This is expected, as critical tools are more frequently used and exposed. At the same time, our defense maintains the ASR at a relatively low level, although it is no longer 0. Therefore, we consider the defense to remain effective even when critical tools are poisoned. We will include these new results and discuss this more explicit critical-tool poisoning setting as a separate subsection in the revision. We will also open-source our implementation.
> > >
> > > However, we would like to clarify that the main attack studied in this work, namely **cross-tool description poisoning**, is a more stealthy and practical attack setting than directly poisoning critical tools or crafting highly similar malicious tools to hijack tool selection. In particular, it allows the adversary to induce harmful downstream behavior without directly modifying the critical tools themselves, which reflects a more realistic attacker capability in the current MCP ecosystem. We focus on this variant because it can still lead to similarly severe consequences, as demonstrated in Section 3.
> > >
> > > We also view **cross-tool description poisoning** as a distinct attack, different from the more explicit **tool preference attack**. The distinction is analogous to the difference between **direct prompt injection** and **indirect prompt injection**: the former directly modifies the main decision target, while the latter more subtly manipulates the surrounding environment to influence downstream behavior. Similarly, tool preference attacks directly interfere with critical tool selection, whereas cross-tool description poisoning more subtly manipulates the broader tool lists. Building on this, we further propose a corresponding system-level defense that aims to ensure security while preserving model utility.
> > >
> > > In summary, we sincerely thank the reviewer for this valuable suggestion, which can significantly enrich and strengthen the paper. We will clarify the threat model and discuss these results in the corresponding sections of the revision.

---

### Decision · Program_Chairs · 2026-04-30

**Decision:**

Accept (regular)

**Comment:**

The reviewers’ final scores for this paper are 3, 3, 4, and 5. Reviewer J1Ay (score: 3) put forward three questions, which the authors addressed in a two-round rebuttal that I believe sufficiently resolves these concerns. Reviewer Lgey (score: 3) accepted the authors’ rebuttal and indicated an intention to increase their score. After reviewing all rebuttal materials, I find that most of the reviewers’ concerns have been adequately addressed. I therefore recommend acceptance as an AC.